# Mothers' perspectives of the barriers and facilitators to reducing young children's screen time during COVID-19: A reddit content analysis

**Leann Blake**[1], **Patricia Tucker**[2], **Leigh M. Vanderloo**[2,3]*

**1** School of Health Studies, Western University, London, Canada, **2** School of Occupational Therapy, Western University, London, Canada, **3** ParticipACTION, Toronto, Canada

* lvande32@uwo.ca

**Data Availability Statement:** The data described and analyzed in this article are openly available on from Western University's Institutional Open-Access Repository (Borealis): https://borealisdata.

## Abstract

Screen time for children under 5 is associated with various health risks. Amidst the COVID-19 pandemic, screen use among young children increased significantly. Mothers were more likely than fathers to be the primary caregivers and disproportionally assumed the responsibility of monitoring their children's screen time. Several studies have examined children's screen use throughout the pandemic; however, few have addressed mothers' experiences. Therefore, the current study aimed to investigate mothers' perceptions regarding the barriers and facilitators faced when trying to reduce their child's pandemic screen time, as expressed on Reddit (a social media platform for anonymous discussion and information sharing). Two subreddit forums targeted toward mothers, "mommit" and "beyondthebump," with 646,000 and 554,000 users, respectively, were examined. Posts were collected using related search terms and screened for inclusion by three independent researchers. Inductive thematic content analysis was leveraged to identify themes. In total, 582 posts were reviewed from March 14th, 2020, to August 31st, 2022. Qualitative analysis yielded 5 themes; 6 barriers and 2 facilitators were derived from themes and/or subthemes, where applicable. Results suggest that mothers faced barriers when trying to reduce their child's screen time, including their competing work and in-home obligations, using screens to occupy their child during travel, child screen use with other caregivers, offering their child screen time while they needed rest, pandemic changes in routine, and using screens to encourage their child to engage in necessary behaviours. However, facilitating factors, including advice received from other mothers on how to reduce their child's screen time and the sharing of non-screen alternatives supported mothers in lowering their children's screen time. These results are important for future interventions, which may utilize the conclusions of this study to address what mothers perceive to be helping or hindering them, thus empowering mothers to successfully limit their children's screen time.

ca/dataset.xhtml?persistentId=doi:10.5683/SP3/
RLC7FO.

**Funding:** The author(s) received no specific
funding for this work.

**Competing interests:** The authors have declared
that no competing interests exist.

## Introduction

Excessive screen time during early childhood may adversely impact physical, cognitive, and emotional development outcomes [1]. For young children (<5 years), excess screen use is associated with anti-social behaviours, poorer self-regulation and attention difficulties, cognitive and language delays, and obesity that may persist into adulthood [2–5]. To minimize potential health risks, the Canadian Paediatric Society (CPS) does not recommend screen time for children under 2 years of age, apart from video-chatting with caring adults, and suggests limiting screen time to less than 1 hour per day for 2-to-5-year-olds [1]. In contrast to the wealth of research highlighting the negative impacts screen time can have on young children, it is also important to acknowledge that there are certain circumstances in which screen time may be positive for children, particularly during the pandemic. Two specific screen activities that may benefit young children include video-chats to foster social connection with distanced loved ones, and virtual story times, which could help with language development and the establishment of routines [1].

Public health measures, such as social distancing and stay-at-home orders, were introduced to help reduce the spread of the COVID-19 virus [6]. Consequently, children were expected to learn, socialize, and play remotely at home, resulting in substantial increases in screen time among children under 5 years during this period [7]. Specifically, prior to the COVID-19 pandemic, young children used screens for approximately 1.5 h/day [8]. This number increased to approximately 2.09 h/day, significantly exceeding the recommended screen time for this age group [8, 9].

Parents of young children have considerable control over the activities in which their children engage [10]. Parents may limit screen use by creating "screen-free" zones within the home (i.e., no screens during mealtimes or in the bedroom), promoting non-screen alternatives, and reducing their own screen use to model this behaviour [11]. However, this is not always a manageable task. Pre-pandemic, Jarvis and colleagues [12] found that parents face more challenges when trying to limit their child's screen time than they do in promoting any other health behaviour. This became increasingly difficult during the pandemic due to the absence of children's regular structured routines and a dearth in available alternatives to screen time [13].

Social media can be used for information or social support during a crisis and played a crucial role in influencing people's health behaviours during COVID-19 [14, 15]. In response to mothers' increased responsibility of managing their children's pandemic health behaviours, including screen time, many leveraged social media to connect with other mothers for support, to generate activity ideas, and to vent their frustrations [16]. One such platform, Reddit, is a free social discussion website allows users to exchange with information and post content on forums designed around specific topics called *subreddits* [17]. These subreddits form communities of users that share common interests or contexts, including motherhood and child rearing [18]. Reddit is valuable for qualitative research for several reasons. First, it is the 18th most popular website in the world, giving rise to large amounts of data [17]. Additionally, posts on Reddit are publicly accessible, and users organize themselves into communities based on specific interests, which allows researchers to easily identify and study eligible participants [17]. Deeper textual analysis is also made possible on Reddit as it lacks character restriction, unlike other social media platforms such as Twitter, which has a 280-character limit that restricts the amount of information users can share with their followers [19]. In addition, Reddit is a text-based platform, in contrast to other popular social media platforms like Instagram, which are photo-based, and focus primarily on users sharing images taken in their daily life with their followers or the public, limiting amount of written, expressive content that may be used for

analysis [20]. Moreover, Reddit's anonymity allows users to share their feelings openly, potentially reducing self-report bias and increasing opportunities for self-disclosure [17]. Previous research on mental health has indicated that Reddit is a valuable source of data for qualitative analysis due to the willingness of anonymous users to express their experiences despite stigma or apprehension that may arise when disclosing their honest thoughts and feelings to mental health professionals and/or friends [21]. Since mothers of young children may experience guilt or shame when sharing the challenges they face as a parent, Reddit offers a safe outlet to be open about their experiences without fear of judgment from others [22, 23].

Numerous quantitative studies have explored children's screen use throughout the pandemic, but many lack insight regarding how to support caregivers to mitigate this challenge [7, 24, 25] In addition, there is a lack of qualitative research that has examined mothers' impressions of their efforts to limit their young children's screen use during COVID-19. Eyler and colleagues [26] explored parents' perceptions regarding their children's physical activity and screen time during the pandemic. However, this study was not specific to mothers' perspectives (parent who is typically reported as being primarily responsible for childcare [27]) and centred on the screen behaviours of children between 5–12 years [26]. Without understanding what mothers perceive to aid or impede them in reducing their child's screen time, future interventions may fail to address critical factors voiced by caregivers who spend the most time with the child. In addition, resources or supports adopted to assist parents based on information specific to 5-to-12-year-olds may not be effective for younger children. While school-aged children increasingly engage in activities independently, the behaviour of those under 5 years is highly dependent on their caregivers as they require constant entertainment and supervision [28]. As such, this study aimed to qualitatively understand mothers' perspectives on the barriers and facilitators to reducing their young children's (0–5 years) screen time during the COVID-19 pandemic by leveraging social posts on Reddit.

## Methods

Posts on Reddit related to screen use were queried and extracted from two subreddit groups for mothers, "Mommit" (https://www.reddit.com/r/Mommit/) and "Beyondthebump" (https://www.reddit.com/r/beyondthebump/), with 688,000 and 570,000 users, respectively. Such subreddits were chosen as they are the two most popular subreddits specifically dedicated to mothers on the Reddit app. Posts were collected using the search terms "screen time," "tablet," "phone," "television," and "TV." Choice of key terms were informed by the literature and expert group (author) discussion. These search terms were used to obtain the initial repertoire of posts discussing the topic of screen use before screening for inclusion. Approval from the university's ethics review board was not required as all published data were in the public domain; there was no waiver of consent. Collection and use of Reddit posts complies with the Reddit user terms and conditions agreement. The data described and analyzed in this article are openly available on from the authors' institutional open-access repository: https://borealisdata.ca/dataset.xhtml?persistentId=doi:10.5683/SP3/RLC7FO.

### Inclusion criteria

An exploratory, data-driven, content analysis was employed, meaning there were no preconceived hypotheses regarding what would be discussed or shared in the posts [29]. Instead, posts were chosen based on the following inclusion criteria, they were: 1. written in English 2. written by someone who identifies as a mother, 3. discussing their young child's (0–5 years) screen use, and 4. posted between March 14th, 2020, and August 31st, 2022. These dates represent the time between the initial public-school closures in Ontario and the formal commitment

by the Ontario government to keep students in school beginning September 2022, including the reintroduction of extracurricular activities [30, 31]. Three independent researchers (L.B, A. H., and A.L.) screened each post for inclusion to ensure consensus. Reddit posts that were gathered during data collection that did not meet the three inclusion criteria described above were excluded and not utilized for subsequent analysis. No demographic information was available on Reddit users per the anonymity of the forums.

### Thematic analysis

Following the collection and screening of Reddit posts for inclusion, and in compliance with the platform's user terms and conditions agreement for data analysis, inductive thematic content analysis, as described by Braun and Clarke [32], was employed to code posts and derive emergent themes. Inductive thematic analysis involved reading and re-reading each post, systematically identifying and naming the aspect related to screen use, searching for patterns, and organizing the data into themes and subthemes [33]. Data analysis was completed manually. One researcher (L.B) individually coded each post and devised emergent themes, reviewing and revising them to ensure intra-coder reliability. Two additional independent researchers (L.M.V. and P.T.) reviewed the coded themes for agreement, and slight changes were made to improve the conciseness and clarity of themes.

Themes were presented with representative quotations and themes and subthemes were classified into barriers and facilitators. Barriers were defined as factors that inhibited mothers' ability to limit their children's screen use, and facilitators were factors that supported or empowered mothers in this pursuit. A mind map was created to visually present barriers in a user-friendly manner.

## Results

The initial search yielded 582 posts uploaded from March 14th to August 31st, 2022. Of these posts, 157 were excluded as they did not meet the inclusion criteria, leaving 425 unique posts to be analyzed. Following analysis, five themes were clear regarding mother's perspectives on their children's pandemic screen time: 1. mothers' alternative obligations creating an increase in child's screen use; 2. request for advice/tips on how to reduce child's screen time or looking to receive or give; 3. offering child more screen time to allow themselves to rest or recover, 4. compromising with child and allowing increased screen use, and 5. Mother's screen use creating background or secondary screen exposure for the child. Within these five themes, 17 subthemes (three in theme one, seven in theme two, three in theme three, two in theme four, two in theme five) were identified which are presented in Table 1. To ease with interpretation, several themes and subthemes were organized into barriers and facilitators.

### Theme 1: Mothers' alternative obligations

In the first theme, mothers' employment, school, or in-home obligations constrained their ability to occupy their children. Mothers felt overwhelmed and wanted to perform well in work or school and complete house chores (e.g., preparing meals and cleaning), leading them to rely on screen use to keep their children entertained during their competing demands. Or, as some mothers expressed, they used screens as "babysitters" for the child while they completed other tasks.

## Theme 2: Asking for advice/tips and support from or venting to other mothers

Second, mothers utilized the Reddit platform to obtain informational support and emotional or peer support (e.g., encouraging mothers to be kinder to themselves, providing relief that other mothers were facing similar screen related challenges). Within this theme, seven sub-themes were identified, which include the following: seeking tips on how to reduce screen time from other mothers (e.g., strategy, rules, routines, technical advice); inquiring about alternatives to screen time; seeking advice to occupy their child during travel without relying on screen use; normalization of screen time amongst mothers, encouraging others to avoid being hard on themselves for their child's screen time; inquiring if a behaviour is bad for their child or about nuances of risk for screen time (e.g., risk of communication applications like Face-Time); expressing concern about child's screen time obtained with another caregiver; and venting about pressure or guilt felt regarding their child's screen use. Mothers broadly understood the importance of limiting screen time for their child's health. For instance, many referenced how screen use could cause damage to their children's eyesight or that it is associated with lack of attention or learning difficulties. However, several mothers wanted more information regarding the type of screens associated with the most significant health risks for their child or whether certain content or purposes of screen time "counted" towards their daily limit. For example, some mothers inquired whether playing educational games or video chatting with grandparents were as harmful to children as other forms of screen use. In addition, many mothers requested guidance to change their child's screen behaviour or to obtain reassurance from others who may be experiencing similar challenges with their children. Finally, mothers expressed feeling pressured to reduce their child's screen time because of the abundance of messaging on social media or from their peers or family members regarding the health risks associated with screen time for children. Such pressure perceived by mothers was often accompanied by feelings of guilt for the mother when they did allow screen time for their child.

**Table 1. Emerging themes and subthemes from Reddit posts made by self-identifying mothers regarding their young child's screen time.**

| Theme | Subtheme | Barrier or Facilitator | Supporting Quotes |
|---|---|---|---|
| **Theme 1: Mothers' other obligations creating an increase in child's screen use.** | 1.1 Mother work obligations (e.g., working from home or frontline worker). (n = 11) | Barrier | "Initially working with my baby was just survival mode. I did whatever kept him happy and calm enough to do work. Unfortunately, the answer is mostly watching Sesame Street for a couple hours a day while he plays in his jumper or activity seat or hangs out on my lap." |
| | | | "I am so grateful for The Good Dinosaur on Disney+ [on-screen TV show], using TGD as a babysitter has provided [me with] time for meal making, online grocery shopping, WFH [work from home], and numerous other activities." |
| | 1.2 Mother doing online school. (n = 4) | | "We try to keep him away from screens completely, but I'm nervous that with school we might fall into allowing it more easily." |
| | | | "Before all of this [school], I was really strict on-screen time. . .I've tried for the past few weeks to stick to that and just do all of my homework at night but it's killing me staying up till midnight and then waking up at 6 with my little girl." |
| | 1.3 To occupy the child while Mom attends to responsibilities in the home (e.g., cooking, cleaning). (n = 18) | | "I try to limit screen time, so I save that for when I'm cooking and cleaning." |
| | | | "I will wake my daughter up and have her come to the living room to watch tv while I get the boys breakfast." |

*(Continued)*

**Table 1.** (Continued)

| Theme | Subtheme | Barrier or Facilitator | Supporting Quotes |
|---|---|---|---|
| **Theme 2: Asking for advice/tips from other mothers on how to reduce screen time or looking to vent, share insight, receive, or give support from community of mothers.** | 2.1 Advice/tips on how to reduce screen time from other mothers (e.g., strategy, rules, routines, technical advice). (n = 49) | Facilitator | "What measures can I put in place that limits screen time to a reasonable level and reduced incessant asking for screens?" |
| | | | "Is there an app that makes it possible for me to set screen time [limits] for phones and tablets to kids using just my own phone (of course apps will on their devices will have to be pre-installed)?" |
| | 2.2 Looking for or sharing options/alternatives to screen time (non-screen alternatives, may be technical). (n = 21) | Facilitator | "Hey hey! If you have kiddos around and need something for them to do that's not screen time or sleeping. Check out Stoopkid Stories Podcast!" |
| | | | "My husband usually fills this time with tv, movies, video games, or phone videos and I'm trying to find a better alternative that we could all enjoy and wouldn't involve 2 hrs of screen time. Although I am okay with occasional movie watching before bed, I do not want it to be an everyday part of our routine." |
| | 2.3 Travel concerns regarding screen time. (n = 31) | Barrier | "My girl is generally pretty happy in the car, but she definitely starts getting cranky around 40 minutes. We can time it to coincide with her nap, but that will only buy us 1–2 hours if we're lucky. Is it time to invest in an iPad to strap to the headrest? I'm not above screen time." |
| | | | "I'm going to be travelling with my 17m old next month and need tips on how to survive the 12h40m plane journey! I've researched toys, screen time, stickers, fav food etc. but how do I make sure she sits during take-off and landing?" |
| | 2.4 Normalization of screen time, encouraging mothers to be avoid being hard on themselves for their child's screen time. (n = 9) | | "Screen time is a very divided topic. There's a lot of shaming around it too. Obviously if your child has become unhealthily addicted, something needs to be done. Never feel bad if you need to distract your kid with a screen, or if they are allowed to have screen time." |
| | | | "It's about balance for us. They have some screen time, but also always get outside time to explore and play in the dirt." |
| | 2.5 Inquiring if a behaviour is bad for their child, asking for other mother's opinions on their child's screen time, or inquiring about nuances of risk for screen time (e.g., risk of secondary/background TV, asking about risk associated with certain amounts of screen time). (n = 46) | | "Just out of curiosity, what do you all consider as "screen time" when discussing what should be limited for young babies? Do you consider anything that involves any kind of screen (tv, movie, phone, computer etc.) or is it specific to content?" |
| | | | "It's [TV] one of the things my 10-week-old loves to stare at, but is it any different than staring at windows and interesting lamps?" |
| | 2.6 Concerns about high levels of screen use with another caregiver. (n = 13) | Barrier | "My husband has a short temper and instead of playing with our son or taking him outside, he gives him screen time half the time so he can play video games." |
| | | | "Which brings me to my next grievance—when they [grandparents] do 'babysit' they park him in front of screens." |
| | 2.7 Venting about pressure or guilt felt regarding their child's screen use (e.g., feeling bad for being unsuccessful in their pursuit to limit screen time). (n = 11) | | "The guilt just eats me alive. I scroll through social media and see so many moms who look like they are doing so much better than me, and I read so many comments that detail how horrible television is to little kids, especially under the age of two, and how you should never ever let them watch anything or play with your phone or interact with screens at all." |
| | | | "I feel like a horrible mother. I feel guilty because I stick my kids in front of a screen for hours on end." |

(*Continued*)

**Table 1.** (Continued)

| Theme | Subtheme | Barrier or Facilitator | Supporting Quotes |
|---|---|---|---|
| **Theme 3: Mothers allowing child more screen time to have a break or recover from illness.** | 2.7 Venting about pressure or guilt felt regarding their child's screen use (e.g., feeling bad for being unsuccessful in their pursuit to limit screen time). (n = 11) | Barrier | "We do screen time at my house. I am 7 months pregnant with our second baby, so my toddler gets to watch a movie almost every day." |
| | | | "Since I'm trying to break our TV habit, any suggestions for keeping toddler entertained and safe while I put my daughter down, 3 times a day? He'll be 2 in a few weeks, so his attention span is still pretty short." |
| | 3.2 Mother sick or injured (e.g., concussion, COVID-19, flu). (n = 23) | | "Don't generally like too much screen time for my 13-month-old daughter. . . but I am so sick with tonsillitis, and I just don't have the energy to do anything else but lay sweating on the sofa and pat her head while she watches moana." |
| | | | "I have been very against allowing any screen time for our baby. She is 4 months old. I am on day three of Covid and I just can't. I gave in and let her watch cartoons so I could just have some rest." |
| | 3.3 Mental health (e.g., exhaustion, depression, anxiety). (n = 40) | | "I've depended on the screen [for child] a lot while dealing with PPD [PostPartum Depression]." |
| | | | "A couple days ago I had therapy and didn't want her interrupting so for the first time in her life I let a screen be her babysitter." |
| | | | "She's [child] watching the screen when I do my daily workout." |
| **Theme 4: Mothers compromising with child and allowing increased screen use.** | 4.1 Pandemic changes in activities or routine resulting in more screen time. (n = 20) | Barrier | "Since we can't go anywhere, I've been letting her watch as much TV as she wants. I felt guilty about it, but I also didn't have much energy to entertain her all day." |
| | | | "You better believe I let my almost 3-year-old have screen time! It's a pandemic, we can't go anywhere, of course I do." |
| | 4.2 Screen use to soothe/calm child or to encourage child to engage in necessary behaviours (e.g., eating, bathing, cutting nails). (n = 42) | Barrier | "Turning tv back on the minute we leave the room so she can watch cartoons (we limit screen time to 10 minutes of sesame street only if she's super fussy)." |
| | | | "My 18-month-old is tantruming more and more, no surprise, but every once in a while it takes a turn and he then feels impossible to calm down. . .Literally the only thing that works seems to be tv, which I do not want to become a thing." |
| **Theme 5: Background or secondary screen exposure for child.** | 5.1 Mother watching screen device while holding child or while child played in the same room to occupy mother. (n = 27) | | "I will never be on leave again just me and the baby. Soaking in the snuggles and contact naps. Binge watching TV all day while we cycle through feedings, tummy time, and naps." |
| | | | "I like to keep the tv on in the background throughout the day. My son is 1 and he doesn't watch it, but I've read that it can impact language. Anyone here played tv all the time? How did your kiddos turn out?" |
| | 5.2 TV or other screen exposure while child sleeps or feeds. (n = 23) | | "It was about 12:30 in the morning and me and my husband were sitting on the couch watching TV while I breast-feed the baby." |
| | | | "We pretty much watch TV all night while we feed him [newborn] and while he sleeps in the bassinet next to us on the couch." |

Note. The number of posts reflected in this table (n = 400) does not equal the total number of posts that met the inclusion criteria and were coded (n = 425). Twenty-five posts did not fit into the five themes above, and insufficient patterns were identified to warrant creation of new themes.

## Theme 3: Mothers needing a break or to recover from illness

Third, mothers offered their child more screen time to allow themselves to rest or recover. Three subthemes included the mother being pregnant with another child or being over-whelmed with multiple children; sick or injured (e.g., COVID-19, concussion, tonsilitis); or

the mother offering child screen time to take a break for their mental health (e.g., exhaustion, depression, anxiety) or to engage in self-care activities (e.g., use screen as babysitter). When mothers felt tired from entertaining multiple children, they used screen devices as a tool to occupy their children of different ages (e.g., infants and toddlers). Mothers also used their child's screen time as an opportunity for themselves to take a break or engage in an activity for their own mental or physical health benefit (e.g., psychotherapy, making themselves a meal, exercising).

### Theme 4: Mothers compromising with child

In the fourth theme, mothers compromised with their children and allowed increased screen time. The two subthemes in this category included pandemic changes to routine (i.e., limited alternative activities due to stay-at-home orders and social distancing measures) and the need to soothe or calm the child or encourage the child to engage in necessary behaviours (e.g., eating, hygienic practices). Posts included in this theme indicated that mothers were conscious of the screen time their children obtained and often tried to limit screen exposure. However, many mothers shared that they loosened screen time rules for their children during the pandemic. Since public health restrictions resulted in the cancellation of their child's usual stimulating or fun activities, mothers offered their child screen time to entertain them or because it was an activity the child could engage in independently. In addition, several mothers expressed that they let their child use screens to incentivize the child to engage in behaviours perceived as non-negotiable (e.g., eating, bathing, trimming nails) or to calm the child when they were restless, fussy, or throwing a tantrum.

### Theme 5: Background or secondary screen exposure for child

Finally, the fifth theme highlighted that mothers' personal use of personal screen devices oftentimes resulted in background or secondary screen exposure for their child. Two subthemes in this category included mothers' use of screens to occupy themselves while holding their child or while the child played in the same room and watching screens while their child slept or fed. Mothers explained their use of screens for background noise and many sought information regarding the risks of secondary screen usage in children (i.e., wondering if secondary screen exposure was of concern to the child's health).

### Mothers' perspectives on barriers and facilitators to screen time

From the themes identified in the reddit post analysis, six were labelled as barriers and two facilitators to screen use in young children. The six barriers that mothers identified to challenged their ability to limit their young child's screen time were: mothers' work, school, or homecare obligations; adoption of screens to occupy the child during road or airplane travel; high use of screen time with other caregivers; mothers offering child more screen time to allow themselves to rest or recover (e.g., self-care practices to reduce exhaustion, mother unable to entertain child while sick with COVID-19); pandemic changes to routine; and using screens to calm or soothe the child or to encourage the child to engage in necessary behaviours (e.g., eating, bathing). A visualization of the barriers is presented in Fig 1.

In contrast to the barriers listed above, mothers also identified factors that supported or empowered them to reduce their young child's screen time. The two facilitators included receiving advice on how to limit their child's screen use, such as sharing technical advice or rules or routines used in other households, and learning ideas for non-screen alternatives like audiobooks, podcasts, or board games. Mothers were able to learn what strategies or activities were found to be successful in other households. For example, many

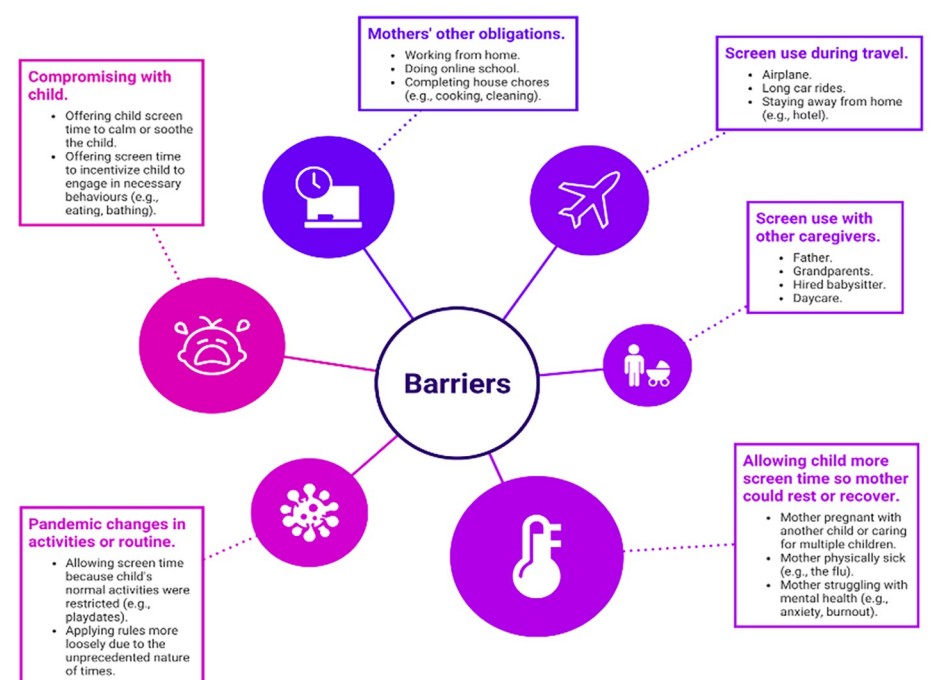

**Fig 1. Mothers' perceived barriers to reducing young children's screen time during the COVID-19 pandemic.**
*Note*: The size of the circles represents the number of posts (i.e., magnitude) discussing each barrier.

received technical advice on how to create a time restraint on their child's device to ensure they were not exceeding their daily screen limit. The facilitators are presented visually in Fig 2.

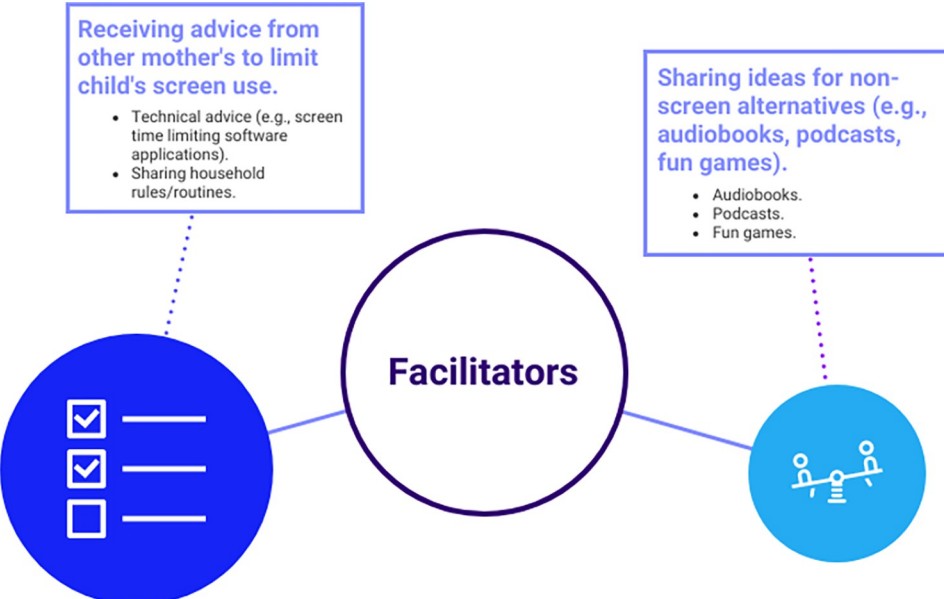

**Fig 2. Mothers' perceived facilitators to reducing young children's screen time during the COVID-19 pandemic.**
*Note*: The size of the circles represents the number of posts (i.e., magnitude) discussing each facilitator.

## Discussion

Amidst the COVID-19 pandemic, public health measures sequestered children to their homes, and resultingly, led to a surge in screen use reported by this young population [9, 34]. Qualitative content analysis of anonymous Reddit posts made by self-identified mothers provided insight into the factors that aided or hindered them in reducing their child's screen time. Results indicate that mothers found it challenging to balance school, work, or household responsibilities and rest when exhausted or sick without relying on their child's use of screens. As a result of adjustments in their child's routine due to COVID-19 public health measures or, because their child was restless or fussy, mothers also expressed the need to compromise and allow their child more screen time. Moreover, they sought opinions or ideas from other caregivers for non-screen-based alternatives or leveraged the platform as a space to vent and share experiences.

Six barriers mothers faced when trying to limit their child's screen time were identified from the five themes and 17 subthemes that emerged from this study. Additional themes and subthemes may have also contributed to increased child screen use, despite not being identified as distinct barriers. For example, posts demonstrated the normalization of screen time among mothers; mothers received reassurance from others that screen time was normal for children and that they should not feel bad about allowing it. While such reassurance may not have caused mothers to increase their child's screen time, it is possible that this influence resulted in mothers no longer feeling like they needed to actively reduce it. Moreover, mothers shared that they often watched TV with their child in the same room or used their phone while their child was on their lap. Thus, while not definitively a factor that prevented mothers from reducing their child's screen time, this theme suggests that mothers' personal screen use might subject their children to increased secondary or background screen time.

Facilitators that supported or empowered mothers to reduce or limit their child's screen use were also identified. These factors included sharing advice on how to limit screen time with other mothers (e.g., strategies, rules, techniques) and sharing examples of alternatives to screen use. A significant number of posts fell within these two subthemes which may indicate that social media platforms such as Reddit can positively support mothers in their quest to monitor and decrease their child's screen use. As described previously, mothers of young children may experience negative emotions of guilt or shame when sharing their personal parenting challenges with others, and Reddit offers an opportunity to share their experiences without fear of judgment [22, 23]. Many mothers explained that they were surrounded by messages from the media, family members or peers about the ills of child screen time and the recommended screen exposure for their child. Such widespread awareness and sharing of the risks associated with screen time could be perceived to help motivate mothers to take action to limit their child's screen use. However, most mothers were already aware of some of the health risks associated with screen use and felt they were trying their best to limit screen time as it was. Thus, receiving continuous input from others often caused them to feel guilty or overwhelmed. The individuals or media sources expressing these messages regarding child screen time may have meant to help mothers. However, such input may be perceived by mothers as shaming those who allow screen time, which can adversely influence their mental health and subsequently impact the child [35].

Future interventions designed to reduce the screen use of young children may be informed by the facilitators that help mothers reduce screen time uncovered in the current study, specifically, the sharing of information and advice amongst mothers on how to limit screen time and sharing examples of alternatives to screen use. For example, such interventions may be public health units or daycare/school communities designing an anonymous virtual program for

parents to share tips and tricks. Due to the anonymous nature, this program could allow parents to feel comfortable and not fear judgement, asking questions and seeking advice from their peers with children the same age. However, in contrast to Reddit, such a platform could have the additional benefit of being specific to parents in one's local area and daycare or school community. Regarding the discussion on how to support mothers with reducing screen time in the future, it is also essential to consider that, as identified in Theme 1, mothers frequently expressed their need to offer their child screens to afford themselves time to attend to their other obligations (e.g., schoolwork, cooking, and cleaning). Thus, screen time is utilized as a tool for mothers to manage their competing roles. This underlines the point that mothers often bear many responsibilities within their families, whether to themselves, their children, or the whole family unit. As such, future interventions implemented to reduce screen use amongst young children may also be created to mitigate maternal overwork or burnout or to offer mothers an alternative outlet to screen time to occupy or 'babysit' children while they attend to other tasks. Pre-pandemic, researchers examined the barriers and facilitators to reducing young children's screen time. Barriers identified in past research have included the ubiquitous nature of screens (e.g., at home, school, daycare), screen use within children's routine, the influence of other caregivers, the perceived addictive nature of screens, child enjoyment, lack of social network, weather, perceived health benefits, parent modelling or co-viewing, screen used as a babysitter, influence of other children, and parents' other responsibilities (e.g., working, caring for elderly parents) [36, 37]. Compared to the results of the current study, some consistent barriers were noted, such as screens being embedded in the child's routine, high child screen use with other caregivers, using screens as a babysitter, and other commitments or obligations for the parent. Regarding routines, the current study results focused specifically on increased screen use practices adopted due to the pandemic. Moreover, using screens as a babysitter was reflected in two themes: mothers used screens to occupy their children while they attended to their other obligations or when they took a moment to rest or engage in self-care practices). Parent modelling or co-viewing (i.e., the child observing parent screen use or engaging in screens with the parent), a barrier identified in pre-pandemic research, is similar to theme five in the current study. In this theme, mothers described their children being exposed to background or secondary screen time. While mothers did not mention mutually enjoying screens with their child (i.e., co-viewing), many did explain that their child was often in the room while they were using a screen, such as watching television. During these instances, the mother may be modelling screen use for the child, and although the child may not be attentively watching the screen, they may be glancing periodically, as was noted by some mothers.

Less surprising is that mothers' mental and physical health emerged as a new factor that limited their ability to reduce their child's screen time during the pandemic. Understandably, many posts regarding mothers' physical health influencing children's screen time were related to having COVID-19. Moreover, a study examining maternal mental health during the pandemic found that most mothers of young children experienced a decline in at least one aspect of their mental health [38]. Therefore, it is comprehensible that maternal mental health emerged as a new barrier to limiting young children's screen time. When mothers experienced anxiety, depression, exhaustion, or burnout, they often relied on their child's use of digital devices while they took a break to relax or rest, rather than having to entertain their children themselves. For example, mothers expressed allowing their children to watch TV while they attended a virtual psychotherapy session or engaged in their favourite exercise routine for self-care. This finding demonstrates that mothers' mental health may influence children's health behaviours, specifically screen time. Such information may inform future efforts that seek to reduce young children's screen time, including interventions that focus on promoting

maternal mental health, particularly during precarious events such as pandemic that can profoundly affect psychological well-being [39].

Lastly, mothers' use of screens to soothe or calm their children or to encourage necessary behaviours (e.g., hygiene practices, eating) emerged as a new barrier not identified in pre-pandemic qualitative research. Mothers expressed that they offered their child more screen time if the child was acting fussy (i.e., whining, challenging to soothe), throwing a tantrum (i.e., extreme anger, upset, crying) or incentivize the child to eat or cooperate during hygienic practices (i.e., nail trimming or bathing). This finding is intriguing, as Raghunathan and colleagues [40] found that during the COVID-19 pandemic, children demonstrated a significant decline in self-regulation ability. Lower self-regulation was indicated by children showing poorer concentration, task engagement and persistence, and greater impulsivity than they did before the pandemic [40]. A similar finding was noted by Katzman [41], who attributed the notable decline in emotional and self-regulation abilities in young children during the pandemic to their lack of social interaction and increased screen use. Lower capacity for children to monitor their behaviour, de-escalate heightened emotions, or cope with stress [42], which are characteristics of child self-regulation, may have resulted in mothers relying more on screens to calm their children. In addition, since high levels of screen use may decrease a child's self-regulation ability [5], it is necessary to consider how the use of screens to calm or soothe a child may not be helping a child regulate their behaviour or emotions. While screens may temporarily calm a child, as described by mothers, screens may be contributing to a decrease in children's ability to regulate their emotions or behaviour, potentially resulting in increased reliance on screens in the future.

Earlier research examining parents' perceptions of their child's screen use indicated that some believed screens offered health benefits to their children, such as allowing them to relax or regulating their sleep patterns [37]. In contrast, posts related to screen time in the current study indicated that most mothers were aware of some potential risks associated with child screen time. Despite being aware of these associations, several mothers wanted more information regarding the nuances of the risks involved, as seen in Theme 5 and Subtheme 2.5 in this study. For example, many were confused regarding which screens "counted" toward their child's daily screen time and wanted to know the implications of background or secondary screen viewing and specific communication apps like FaceTime. This finding may indicate an existing knowledge gap. Public health messaging may have effectively spread awareness to mothers regarding child screen time risks and time recommendations; however, mothers desire a greater understanding of the types of screen exposure or content associated with the most significant health risks for their children.

Furthermore, widespread messaging regarding the risks of screen time and the influence of the media, peers, or other family members led several mothers to feel guilty when offering screen time to their children, as seen in Subtheme 2.7. With this understanding, it may be beneficial for future public health efforts to move away from absolute thinking or rigid time restrictions. Instead, translating practical strategies that consider our current context and reliance on screen use could help mothers promote their child's health and well-being within an increasingly virtual world. Recently, the CPS updated their screen time guidelines for children under five, moving away from rigid screen limits [1]. The CPS recommends avoiding screen time one hour before bed for children under five, prioritizing age-appropriate, educational, and interactive screen use, and prioritizing family media use over solitary screen use for children [1]. While such recommendations may allow mothers to make informed decisions regarding their children's screen time, it is imperative that such information is properly disseminated and made accessible to all mothers.

## Strengths and limitations

The strengths of the current study regarding the trustworthiness of the data collection and analysis can be assessed using Lincoln and Guba's evaluative criteria for qualitative research [43]. Lincoln and Guba posit that the trustworthiness of qualitative research includes establishing transferability, credibility, dependability, and confirmability [43]. Transferability is achieved by offering readers descriptions of how the findings may apply to other contexts [44]. In the current study, this was done in the discussion by offering suggestions on how future interventions could utilize the results of the present study to minimize screen time for young children. Credibility refers to how congruent the findings are with reality [44]. In qualitative research, this can be achieved by investigator triangulation, whereby multiple researchers complete comparative analyses of the data [44]. This was undertaken in the current study as independent researchers were involved in screening posts for inclusion and establishing themes. Dependability encompasses how confidently readers can rely on the findings [44]. One mechanism to achieve dependability is through peer debriefing throughout the analysis process [44]. Moreover, confirmability refers to getting as close to objective reality as qualitative research can get [44]. Confirmability in qualitative research is assured when data are checked and rechecked throughout data collection and analysis to ensure results would likely be repeatable by others [44]. Both dependability and confirmability were achieved in the current study through open discussion amongst authors and research assistants to achieve consensus and mutual understanding regarding which posts would fit the inclusion criteria, which themes emerged, and how posts would be classified within each theme. In addition to the trustworthiness of data and analysis of the current study, a strength of this study is that anonymous discussion forums, like Reddit, allow people to engage in honest communication that may be difficult or sensitive to discuss in face-to-face settings [45, 46]. This study captured authentic emotions from mothers and offered unique insight into their perceptions regarding their young child's screen use they may not have otherwise shared, perhaps out of embarrassment or shame. However, the utilization of anonymous social media platforms for research does have limitations. Collecting demographic information of participants was not possible. Although population-level data have not been reported for individual subreddits, most users of the Reddit app are from the United States [47]. While it has been indicated that users are relatively representative of the American population, there are differences between the general public and Reddit users [47]. For example, Barthel [47] found that Reddit users were more educated than the general population. Moreover, Hargittai [48] argues that those of higher socioeconomic status are more likely to be on social media; and therefore, their views may be oversampled in big-data research, such as that collected on the Reddit platform. The perspectives captured in the current study may be skewed to represent unique challenges faced by mothers who are more educated and are from a higher socioeconomic background than the general population. In addition, the platform's anonymous nature can affect the content's validity. It is also possible for Reddit users to provide false or dishonest information [49].

## Conclusion

The results of this qualitative work posit that young children's screen time may have been influenced by mothers' mental and physical health experiences during the pandemic. Moreover, reasons why mothers' relied on screen use through the pandemic varied greatly, from soothing or calming their child, to incentivising them to engage in bedtime routines (e.g., bathing, teeth brushing) or other necessary behaviours. As technology becomes increasingly embedded in children's lives, interventions to reduce screen time should be tailored to address the needs voiced by caregivers who spend the most time with their children. Thus, future

efforts to reduce young children's screen time may aim to improve mothers' mental health and well-being, so they can better take action to reduce their child's screen time.

## Acknowledgments

The authors would like to acknowledge and give thanks to research assistant's Ameena Haddara and Aidan Loh for their time and commitment in screening Reddit posts for their inclusion for analysis in the current study.

## Author Contributions

**Conceptualization:** Leann Blake, Patricia Tucker, Leigh M. Vanderloo.

**Data curation:** Leann Blake.

**Formal analysis:** Leann Blake.

**Methodology:** Leann Blake, Patricia Tucker, Leigh M. Vanderloo.

**Writing – original draft:** Leann Blake.

**Writing – review & editing:** Patricia Tucker, Leigh M. Vanderloo.

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
