## [Decision Letter · Decision Letter 0]

28 Sep 2023

PONE-D-23-18207Mothers’ Perspectives of the Barriers and Facilitators to Reducing Young Children's Screen Time During COVID-19: A Reddit Content AnalysisPLOS ONE

Dear Dr. Vanderloo,

Thank you for submitting your manuscript to PLOS ONE. After careful consideration, we feel that it has merit but does not fully meet PLOS ONE’s publication criteria as it currently stands. Therefore, we invite you to submit a revised version of the manuscript that addresses the points raised during the review process.

We look forward to receiving your revised manuscript.

Kind regards,

Fatch Welcome Kalembo, Ph.D

Academic Editor

PLOS ONE

Journal Requirements:

2. In your Methods section, please include additional information about your dataset and ensure that you have included a statement specifying whether the collection and analysis method complied with the terms and conditions for the source of the data.

4. Please include a caption for figure 1. 

**Additional Editor Comments:**

Based on the reviewers’ reports, and my assessment as Editor, I am pleased to inform you that the manuscript is potentially acceptable for publication in PLOS ONE once you have addressed the reviewers’ comments. Also, please consider addressing the following in your revised manuscript:

From the findings of the study, it is clear that the study’s main overarching themes were barriers and facilitators to screen time and the five themes identified fall under these two main themes. The authors could improve the clarity and reporting of the results in this manuscript by presenting the five themes under these two main themes.It is also important for the authors to be clear in terms of the how trustworthiness of data was achieved in this study. Perhaps you could add a paragraph after the data analysis section to explain how the trustworthiness of data was achieved in this study.The discussion section could also be strengthened by a clear description of the implications of study findings for practice and researchThe manuscript also needs some editing before it is published.

Reviewers' comments:

Reviewer's Responses to Questions

**Comments to the Author**

1. Is the manuscript technically sound, and do the data support the conclusions?

Reviewer #1: Yes

Reviewer #2: Yes

Reviewer #3: Yes

2. Has the statistical analysis been performed appropriately and rigorously? 

Reviewer #1: N/A

Reviewer #2: N/A

Reviewer #3: Yes

3. Have the authors made all data underlying the findings in their manuscript fully available?

Reviewer #1: Yes

Reviewer #2: Yes

Reviewer #3: Yes

4. Is the manuscript presented in an intelligible fashion and written in standard English?

Reviewer #1: Yes

Reviewer #2: Yes

Reviewer #3: Yes

5. Review Comments to the Author

Reviewer #1: This is a clearly written and topical paper. Rigorous methods were used, which were clearly explained. Sound recommendations were made based on the findings.

Figure 1 is very effective.

There was a good discussion on how some sub-themes may have contributed to an increase in screen time without necessarily being a barrier.

The authors acknowledged the limitation of not having demographic data, and how these findings may only apply to higher socio-economic groups from the US. This acknowledgement is to be commended.

This paper makes an interesting link between mothers' mental health and wellbeing and ability to limit screen time. This will inform future interventions and policies regarding family wellbeing.

Additional comments:

Line 1 on p6 is repeated at line 6.

Table 1 - theme descriptions are not necessary given the themes are described in text.

Barriers and Facilitators - please be a little clearer that the themes for barriers and facilitators came from the 5 themes and 17 subthemes in table 1. This is made clear in the discussion but was less clear in the body. Currently it is a little confusing.

There are a lot of brackets used. Can these please be reduced. This will require re-writing some sentences.

Reviewer #2: Dear Authors,

Thank you for the opportunity to review this study that looks at barriers and facilitators of screen time of young children.

This study is relevant, not only during the COVID-19 restriction period, but also provides foundation for future recommendations in children's screen time management. Please find below few suggestions to enhance the manuscript:

1. Authors stated no Approval was obtained from relevant Institutional Review Board. Was there a waiver of consent?

2. There was repetition of approval statement in page 6, please check and delete one.

3. How did the authors identify the origin of the post in page 6? How sure are you that all the post during this time period were from Ontario? A clarification will be great for readers. If this is not possible to identify, perhaps, you could include this as a limitation of the study because the demographics information of the posts were not available.

4. Regarding the first coder, perhaps, you may want to include their names or initials in page 6.

RESULTS

5. Perhaps, authors can consider organising these themes into the two broad categories i.e. Barriers and Facilitators. Mapping the themes under these categories can enhance flow and clarity for readers.

Additionally, the themes can be summarised, i.e. short, sweet and precise.

6. You may want to include the theme's label as part of the sub-heading instead of only indicating Theme One, Two etc

7. Figure 1 looks great for the barriers, what about the facilitators? As mentioned above, organising the themes and sub-themes into the two broad categories will make it clearer and cover the research aim.

DATA ANALYSIS

7. Was any application used to support the data analysis or was it done manually?

Reviewer #3: Comments to the Author

This study aimed to qualitatively understand mothers’ perspectives on the barriers and facilitators to reducing their young children’s (0-5 years) screen time during the COVID-19 pandemic. Through an exploratory data-driven (Reddit) content analysis, five themes emerged regarding mothers’ perspectives of screen time: (1) mothers’ other obligations creating an increase in child's screen use; (2) request for advice/tips on how to reduce child’s screen time or looking to receive or give; (3) offering child more screen time to allow themselves to rest or recover; (4) compromising with child and allowing increased screen use; and (5) mother’s screen use creating background or secondary screen exposure for the child. Authors concluded that parents faced new challenges when trying to reduce their child’s pandemic screen time and findings can be utilised to support the development of interventions designed to support mothers to limit their children’s screen time. The manuscript is well written, and findings are clearly articulated. I would accept with manuscript with minor revisions.

General comment:

Introduction

1. The authors provide a comprehensive overview of the research topic but would benefit from a more balanced argument related to the evidence for the implications of screen use on health. Currently, the evidence presented in paragraph 1 emphasises the detrimental impacts of screen use on the health and development in children. Authors should counter these arguments with more recent evidence related to the potential health and developmental benefits, considering the nuances and complexity of screen use. Otherwise, there is an assumption that screen ‘time’ should be limited, including potential types of screen use that might provide benefit. Authors correctly describe some of the complexities related to social media and around guidelines (e.g., video-chatting with caring adults). A broader, more complex picture needs to be presented.

Methods

2. Some justification behind the selection of subreddit groups (Mommit and Beyondthebump) to analyse posts is warranted. Are there others? It would also be helpful to provide some rationale behind why authors selected key terms (e.g., screen time, tablet, phone, TV). Why were such terms considered important?

Results and Discussion

3. Results are presented clearly. The authors are commended for how themes were presented in the Table and Figure.

4. While the Results are comprehensively interpreted in the Discussion, the implications for future research are somewhat abstract. Given the extensive information captured related to the barriers and potential solutions (facilitators) of supporting mother to limit (or manage) their children’s screen use, there is an opportunity to discuss this in the context of intervention and current guidelines. More importantly, future research might also need to consider the feasibility of such solutions. Some expansion on key facilitators in terms of potential application in interventions is needed.

5. Another important point to consider, when discussing the potential facilitators for mothers when managing young children’s screen use, is that screen time is potentially one part of many concerns’ parents have related to their child’s health and development. Indeed, mothers might use screens as a form of “babysitter” to soothe or calm their child’s behaviour which have potential negative impacts on self-regulation; however, a trade off might be necessary if the mother has competing challenges to face, particular if there is no support elsewhere (e.g., cooking dinner, chores etc.). A more balanced argument that considers the ‘parenting chaos’ would also benefit any discussion around the implications for intervention.

Specific comments:

6. The whole manuscript would benefit from a thorough type-editing review to minimise errors. For example, paragraph 5 has an error with indentation. Moreover, in Methods, “Approval from the university’s ethics review board was not required as all published data were in the public domain” is presented twice. Please remove one to avoid replication.

6. PLOS authors have the option to publish the peer review history of their article (what does this mean?). If published, this will include your full peer review and any attached files.

Reviewer #1: **Yes: **Dr. Shelley Gower

Reviewer #2: **Yes: **Esther Adama

Reviewer #3: No

---

## [Author Response · Author response to Decision Letter 0]

19 Feb 2024

JOURNAL REQUIREMENTS:

1. We note your current data availability statement reads as follows:

"The data described and analyzed in this article are openly available on the Reddit platform at https://www.reddit.com/r/Mommit/ and https://www.reddit.com/r/beyondthebump/."

However, Reddit is not an acceptable data sharing platform.

To ensure you submission adheres to the PLOS ONE Data Availability policy (https://journals.plos.org/plosone/s/data-availability), please either upload the minimal anonymized data set required to replicate your study’s findings as Supporting Information, or to a stable public repository. Please also provide all relevant DOIs, URLs, and or/accession numbers required to access your data on the repository. For a list of acceptable repositories, please see https://journals.plos.org/plosone/s/recommended-repositories.

ACTION TAKEN: The data described and analyzed in this article are openly available on from Western University's Institutional Open-Access Repository (Borealis): https://borealisdata.ca/dataset.xhtml?persistentId=doi:10.5683/SP3/RLC7FO.

The above sentence has been included in the first paragraph of the “Methods” section of the paper (page 7).

---

## [Editor Report · Decision Letter 1]

11 Mar 2024

Mothers’ Perspectives of the Barriers and Facilitators to Reducing Young Children's Screen Time During COVID-19: A Reddit Content Analysis

PONE-D-23-18207R1

Dear Dr. Vanderloo,

We’re pleased to inform you that your manuscript has been judged scientifically suitable for publication and will be formally accepted for publication once it meets all outstanding technical requirements.

Kind regards,

Fatch Welcome Kalembo, Ph.D

Academic Editor

PLOS ONE

Additional Editor Comments (optional): Please update the abstract in the editorial system so that it matches the one provided in the revised manuscript.

---

## [Editor Report · Acceptance letter]

18 Mar 2024

PONE-D-23-18207R1 

PLOS ONE

Dear Dr. Vanderloo, 

I'm pleased to inform you that your manuscript has been deemed suitable for publication in PLOS ONE. Congratulations! Your manuscript is now being handed over to our production team.

Kind regards, 

on behalf of

Dr. Fatch Welcome Kalembo 

Academic Editor

PLOS ONE